# Addition of Regular Insulin to Ternary Parenteral Nutrition: A Stability Study

**DOI:** 10.3390/pharmaceutics13040458

**Published:** 2021-03-27

**Authors:** Heloise Henry, Damien Lannoy, Patrice Maboudou, David Seguy, Thierry Dine, Pascal Pigny, Pascal Odou

**Affiliations:** 1ULR 7365–GRITA–Groupe de Recherche sur les Formes Injectables et les Technologies Associées, University of Lille, F-59000 Lille, France; heloise.henry@univ-lille.fr (H.H.); thierry.dine@univ-lille.fr (T.D.); pascal.odou@univ-lille.fr (P.O.); 2Institut de Pharmacie, CHU Lille, F-59000 Lille, France; 3Service de Biochimie Automatisée Protéines, CHU Lille, F-59000 Lille, France; Patrice.MABOUDOU@CHRU-LILLE.FR; 4Service Endocrinologie Diabétologie Maladies Métaboliques et Nutrition, CHU Lille, F-59000 Lille, France; david.seguy@univ-lille.fr; 5U 1286–Infinite–Institute for Translational Research in Inflammation, University of Lille, F-59000 Lille, France; 6Inserm, U 1286, F-59000 Lille, France; 7Laboratoire de Biochimie & Hormonologie, Centre de Biologie Pathologie, CHU Lille, F-59000 Lille, France; Pascal.PIGNY@CHRU-LILLE.FR

**Keywords:** regular insulin, parenteral nutrition solutions, stability, immunoassay

## Abstract

Background: Parenteral nutrition (PN) is a complex medium in which added insulin can become unstable. The aim of this study is, therefore, to evaluate the stability of insulin in PN and to identify influencing factors. Methods: A total of 20 IU/L of regular insulin was added to PN in either glass or Ethylene Vinyl Acetate (EVA) containers. A 24 h stability study was performed via an electrochemiluminescence immunoassay in different media: A ternary PN admixture, separate compartments of the PN bag and a binary admixture. This study was repeated in the absence of zinc, with the addition of serum albumin or tween and with pH adjustment (3.6 or 6.3). Insulin concentration at t time was expressed as a percentage of the initial insulin concentration. Analysis of covariance (ANCOVA) was applied to determine the factors that influence insulin stability. Results: In all PN admixtures, the insulin concentration ratio decreased, stabilising at a 60% and then plateauing after 6 h. At pH 3.6, the ratio was above 90%, while at pH 6.3 it decreased, except in the amino acid solution. ANCOVA (r^2^ = 0.68, *p* = 0.01) identified dextrose and pH as significant factors influencing insulin stability. Conclusion: A low pH level seems to stabilise insulin in PN admixtures. The influence of dextrose content suggests that insulin glycation may influence stability.

## 1. Introduction

Parenteral nutrition (PN) containing a highly concentrated dextrose solution causes hyperglycemia, reported as a metabolic side effect in 50% of diabetic and nondiabetic patients [1,2,3]. This hyperglycemic state has been proven to be harmful to patients by increasing the length of hospital stay, prevalence to infection, and mortality [4]. Biosynthetic short-acting insulin is often either Y-site administered or directly added into the parenteral nutrition admixture (PNA) in order to reduce hyperglycemia [5]. These options are recommended by both the American and European societies of nutrition (ASPEN and ESPEN), but only if the stability of both insulin and the PNA has previously been demonstrated [6,7]. With the Y-site infusion method, the contact time between the PNA and insulin is limited by flow rate (which is frequently high for adult patients) and the common space volume in the plastic tubing. However, this contact time increases considerably when insulin is directly added to the PNA as it can last for 24 h (for continuous administration of PNA) [8]. In vitro data are scarce on short-acting insulin stability in ternary (three-in-one) PN and mainly concern Y-site administration with short-term study periods [9,10,11,12]. Here, we performed a thorough stability study. First, considering that PNA is a complex medium, each compartment of the bag (containing each macronutrient) was tested alone, while we also tested the binary (two-in-one) admixture. Second, several parameters known to potentially influence insulin behaviour were explored: the effect of zinc [13], the nature and concentration of lipid emulsion, and insulin adsorption [14].

The aim of the present work was to assess the in vitro stability of regular insulin—corresponding to human short-acting insulin—in a three-in-one PN admixture via an immunoassay method. Additionally, we also identified which factors influence the in vitro stability of insulin.

## 2. Materials and Methods

### 2.1. Products

The 2-in-1 binary lipid-free and the 3-in-1 ternary PN admixtures were obtained by mixing 2 or 3 chambers of an Olimel^TM^ N7E bag (1.5 L bags, Baxter, Deerfield, FL, USA). This PN product is available in a three-chamber multilayered bag, each chamber containing, respectively, a 35% dextrose solution and calcium (chamber 1); an 11.1% amino acid solution with other electrolytes (sodium, potassium, magnesium, phosphate, acetate and chloride) (chamber 2); a 20% lipid emulsion (chamber 3). Assays were performed on both 3-in-1 and 2-in-1 PN admixtures and for each compartment of the bag. Vitamins (Cernevit^TM^, Baxter, Deerfield, FL, USA) and trace elements (Nutryelt^TM^, Aguettant, Lyon, France) were added—when needed—to the PN admixtures (1 vial of each in 1.5 L of PN). Cernevit^TM^ was reconstituted with 5 mL of 0.9% NaCl before being added to the PN. Isotonic saline serum (0.9% NaCl) was purchased from Baxter. Zinc gluconate (Aguettant, Lyon, France) and other types of injectable lipid emulsions (Medialipide^TM^, Intralipide^TM^, Omegaven^TM^ and Smoflipid^TM^) were used in various PNA conditions. Water for injection, purchased from CDM Lavoisier (Paris, France), was used to dilute the lipid emulsions. Bovine serum albumin (BSA V-fraction, Euromedex, Souffelmeyersheim, France) and tween 20 (Euromedex, Souffelmeyersheim, France) were used in the sorption assay.

Umuline rapide ^TM^ 100 IU/mL (Eli-Lilly, Suresnes, France), a biosynthetic short-acting (regular) insulin was added to each condition tested at a final concentration of 20 IU/L. All assays were performed at daylight exposure and ambient temperature, in two different types of containers: glass flasks and Ethylene Vinyl Acetate (EVA) bags (purchased from Bexen Medical, Hernani, Spain).

### 2.2. Sample Quantification Method: Preparation, Instrumentation and Kit

Sample dilution and quantification were carried out as previously described in the validation of the method, in a similar medium [15]. The advantage of this quantification method is that insulin can be quantified in a complex medium without any extraction step. This method was performed at each sampling time, after a two-step 1/200 dilution in a 4% bovine serum albumin (BSA) phosphate buffer saline (PBS) 0.04 M solution (sodium dihydrogen phosphate monohydrate and sodium phosphate dibasic dodecahydrate, Cooper, Melun, France) [16]. An electrochemiluminescence immunoassay (ECLIA) Insulin Elecsys^TM^ (Roche Diagnostics, Meylan, France), previously validated for insulin quantification in the 3-in-1 PN admixture, was used, in addition to an e601 Cobas instrument (Roche Diagnostics, Mannheim, Germany), which was calibrated and controlled with a Calset insulin^TM^ calibrating kit and Precicontrol Multimarker^TM^ solutions purchased from the same manufacturer (Roche Diagnostics, Mannheim, Germany). Four quality control samples (7.5, 15, 25 and 35 IU/L) were prepared every day, for each tested medium and quantified by ECLIA. Results are provided as insulin concentration in µIU/mL.

### 2.3. Preparation of PN Admixtures Containing Insulin

The stability of regular insulin was first studied in the 3-in-1 PN admixture (Olimel N7E^TM^ supplemented with vitamins (V) and trace elements (TE), named PN_VTE_). The addition of insulin to the PN admixture and transfer of the final mix to both types of containers is detailed in Figure 1.

This preparation protocol was valid for every tested condition (only the PN medium changed for other conditions). PN medium with added insulin was prepared in 3 distinct flasks (F1, F2 and F3), half of which was then transferred to 3 EVA bags (B1, B2 and B3). At defined sampling times (0, 1, 2, 4, 6, 8, 14, 20 and 24 h after insulin addition), one sample was taken from F1, F2, F3, B1, B2 and B3.

Samples were diluted and insulin was quantified as described above. Additionally, the pHs of the samples were measured with a pHmeter (VWR international, Fontenay-sous-Bois, France).

The study was completed by assessing the impact of adsorption (conditioning assay), zinc, PN admixture composition and pH level on insulin stability.

### 2.4. Study of Parameters Able to Influence Insulin Stability

The influence of the following parameters on insulin stability was assessed: interaction between content and container, micro- and macronutrient content, and pH. To reduce the number of assays, all the experiments described below were performed following a simplified experimental protocol: sampling times were limited to 0, 4 and 24 h after insulin addition to the PN admixtures.

#### 2.4.1. Interaction between Content and Container: Preconditioning and Conditioning Assays

Insulin adsorption has been previously described [17,18,19,20,21,22,23]. Our experiments were split into two-parts: Saturation of adsorption sites before the stability test (preconditioning assay) and saturation of adsorption sites during the stability test (conditioning assay, corresponding to the addition of a saturating element directly into the PN admixture during the stability test). The objective was to confirm or reject the hypothesis of insulin adsorption on containers.

##### Preconditioning Assay

BSA and tween 20 were used as saturating elements. A total of 3.3 g/L of solutions of both were prepared in either phosphate buffer or distilled water. Both types of containers were filled with these solutions and incubated for 72 h, after which, they were emptied and air-dried. Thus, 250 mL of ternary PN supplemented with regular insulin at 20 IU/L was put into each container and the stability study was repeated as previously described.

##### Conditioning Assay

BSA and tween 20 were directly added at two concentrations (3.3 or 40.0 g/L) to ternary PN supplemented with 20 IU/L of regular insulin. Each tested container was filled with the global admixture (ternary PN + regular insulin + saturating element), thus, starting the stability study.

#### 2.4.2. Influence of Micro- or Macronutrient Content and pH

##### Influence of Zinc (Micronutrient)

Insulin hexamerisation is influenced by the presence of zinc [24,25]. In this section, the stability study was performed with and without zinc. Considering that Nutryelt^TM^ contains 10 mg of zinc, it was not added to the 3-in-1 PN admixture, whereas Cernevit^TM^ was systematically added. Two conditions were compared: Adding one vial of zinc gluconate, containing 10 mg of zinc, to PN with vitamins (PN_v_) (condition A) or not adding zinc (condition B). This study was performed over four hours.

##### Impact of the PN Medium (Macronutrient Content)

The stability study was repeated, varying the nature and composition of the PN admixture. Olimel^TM^ N7E 3-in-1 PN, as previously described, was used as the reference and compared to other PN media. The composition and pH of nutrient compartments taken from Olimel^TM^ N7E three-chamber bags and their various 3-in-1 and 2-in-1 combinations are detailed in Table 1.

First, experiments dealing with macronutrient content were performed with the PN media described in Table 1.

Commercial 3-in-1 PN admixtures may contain different lipid emulsions. Consequently, in a second phase, the stability study was performed using other lipid emulsions to be compared with that from a bag of Olimel^TM^ N7E used as reference at a 20% concentration, and after diluting with water for injection in some cases (Table 2).

##### Impact of pH

The influence of pH was studied in each previously tested medium (Table 1). In these media, pH was adjusted with either hydrochloric acid (HCl 25%) or sodium hydroxide (NaOH) (Merck Millipore, Molsheim, France) at 2 levels: 3.6 and 6.3. Stability assays were repeated at both pH values.

### 2.5. Data Expression and Statistical Analysis

Insulin concentration at t time (C_t_) was expressed as a percentage (mean ± SD) of the initial insulin concentration (C_0_) in order to graph the stability of insulin over time. For 3-in-1 PNA, a comparison between types of containers (glass flask and EVA bag) was made using Wilcoxon’s rank-sum test with risk set at α = 5%. Insulin concentrations (C_t_) (µIU/mL) were submitted to an analysis of covariance (ANCOVA) to determine parameters influencing insulin stability.

The best kinetic model describing the evolution of insulin concentration over time was determined. Values corresponding to √C_t_, ln(C_t_), C_t_, 1/√C_t_ and 1/C_t_ were calculated, respectively, to explore 0, 0.5, 1, 1.5 and 2 order kinetics. After checking the normality of the sample distribution, an ANCOVA analysis was conducted with quantitative and qualitative covariates (Table 3) to determine a potential relationship between each factor and the evolution of the insulin concentration.

This was followed by an ultimate ANCOVA analysis, with only significant covariates used independently or in interaction.

## 3. Results

### 3.1. Study of Insulin Stability Depending on Conditions

Figure 2 presents the evolution of insulin concentration over time, expressed as a percentage of the initial insulin concentration (C_t_/C_0_ × 100) with a three-in-one PN in both types of containers over a 24 h period. Evolution over time can be divided into two periods: A sudden drop of nearly 30% within the first 8 h, followed by a steady state phase until 24 h. There was no statistical difference according to the type of container (*p* > 0.181 at each tested time).

The evolution in insulin concentration obtained by preconditioning and conditioning assays in three-in-one PN admixture is presented in Figure 3. No difference was observed, regardless of the saturating element or its concentration.

Additionally, no difference was observed between the presence or absence of zinc element over a 4-h period in the three-in-one PN admixture either (Figure 4).

The impact of pH in the interaction with the PN medium is presented in Figure 5.

This figure was obtained from real values of insulin concentration evolution with different PN compositions.

Insulin concentration decreases over time when the composition of PN contains dextrose at pH 6.3, while it remains stable regardless of the medium at pH 3.6.

### 3.2. Identification of Parameters Able to Influence Insulin Stability

A total of 1656 results were analysed by ANCOVA (including all conditions tested).

Among all tested orders for kinetic reaction (0, 0.5, 1, 1.5 and 2), second-order reaction kinetics were the most appropriate, so ANCOVA analysis followed the Equation (1):(1)$$\frac{1}{C_{t}}-\;\frac{1}{C_{0}}=k.t$$

Parameters detailed in Table 4 were selected for the final ANCOVA because of their statistical significance alone or in interaction.

Thus, the coefficient of correlation r^2^ = 0.68 (*p* = 0.01) was obtained for this analysis, following Equation (2) (correspondence of letters is detailed in Table 4):(2)$$\frac{1}{Ct}=\;11.1-0.311\times A\;-\;0.126\times B\;-\;2.518\;\times 10^{-2}\times C-\;1.247\times D\;+\;8.679\times 10^{-2}\times E\;+\;4.021\times 10^{-2}\times F\;-\;2.695\times 10^{-2}\times G\;$$

All other factors have no statistical significance on insulin concentration. ANCOVA analysis results were obtained as normalised coefficients, indicating the relative influence of significant factors. These normalised coefficients are presented in Figure 6.

Time and dextrose content have a negative impact on insulin stability, both when taken independently and also in interaction with pH.

## 4. Discussion

Regular insulin appeared to be unstable after being added to a three-in-one PN admixture, with a sharp 30% decrease taking place within 8 h. Dextrose content, time and pH were the most significant factors that influenced insulin stability. Indeed, a low pH level seemed to stabilise insulin in all PN admixtures, even in the presence of dextrose. This instability could not be accounted for by hexamerisation (because of the zinc present), by the nature and concentration of the lipid emulsion, or by adsorption on the container.

Hyperglycaemia is one of the most frequent side effects of PN and can be limited by adding insulin, which reduces the risk when PN administration has to be stopped, unlike with Y-site infusion or subcutaneous injection. The main difficulty with drug stability studies on PN solutions is that it is impossible to extrapolate the results to several PN admixtures. In the present work, we broke down the complex PN admixture and tested physicochemical parameters both separately and in association.

Regular insulin, set as the standard in its pharmacotherapeutic class, was chosen because of its short-acting pharmacokinetic profile, which fits well with continuous intravenous dextrose intake, corresponding to PN administration. Regular insulin has already been frequently described in clinical practice for patients experiencing hyperglycaemia because of PN, either by dedicated intravenous infusion [26,27,28,29] or directly added to PN admixtures [26,27,29,30,31,32,33,34,35,36]. Only one clinical study of rapid-acting aspart insulin directly added to PN has been described, as compared to subcutaneous detemir administration [37]. For this reason, regular insulin was chosen for this study. No data have been reported on the stability of insulin analogues in PN. These should, however, be investigated in further experiments and potentially compared to our results. Long-acting insulin analogues (i.e., detemir or glargine), on the other hand, are not appropriate for administration in PN admixtures. In clinical practice, these are only administered subcutaneously [26,31,32,33,34,36,37,38].

Insulin concentration is frequently measured by separative methods, such as high-performance liquid chromatography coupled with UV detection [39,40,41] or size exclusion [42]. However, these separative methods cannot be used to quantify insulin in three-in-one PN emulsions without an extraction step that has the potential to modify the structure of the protein, thus, distorting the results. Consequently, an electrochemiluminescent immunoassay was chosen as the quantifying method, ensuring high specificity. Although lipids can be a huge hindrance for insulin quantification [43], provoking interferences (matrix effect) in the quantification step, this analytical issue was overcome by diluting samples before insulin quantification. This allowed us to respect the measurement range of the ECLIA kit and is a real advantage in terms of reliability of results, compared with a previous stability study of insulin in PNA [44]. In that study, the authors did not consider the gap between insulin concentration in PN and the calibration range. Moreover, the main difficulty concerning insulin quantification in PNA is the fact that, in clinical practice, low concentrations of insulin are used (usually corresponding to 10 or 20 U/L, i.e., 1 U for 10 to 20 g of carbohydrates) [36,45]. Although the abovementioned study tested insulin concentrations ranging from 3 to 36 U/L, the authors failed to quantify the 100% reference of insulin concentration right after adding it to the PNA [44] and did not quantify insulin at all in two-in-one media.

Our first stability study of insulin in three-in-one PNA supplemented with vitamins and trace elements showed a significant decrease in insulin concentration in the first few hours of contact. A 10% loss was observed at 2.5 h, demonstrating the instability of insulin in such a complex medium. Because of the biphasic shape of the curve (rapid decrease ending in a plateau), the main hypothesis was: insulin decrease was due to its adsorption on the container. This biphasic shape is represented by a decrease corresponding to the adsorption phase and a plateau corresponding to the time after saturation of all adsorption sites. Adsorption of proteins in containers is a well-known phenomenon, so two different container materials were compared: EVA and glass, which is supposed to be the more inert of the two [46]. Nevertheless, no statistical difference was observed between the materials at this insulin concentration. Moreover, insulin easily adsorbs on the hydrophobic surfaces (i.e., polyvinyl chloride, EVA, glass, polypropylene, polyethylene, polystyrene) [14,18,22,47,48,49,50,51] and the resulting conformational changes led to aggregation [52,53,54], which is described as a biphasic reaction. To explore this hypothesis, preconditioning and conditioning assays were performed. The aim was to saturate the adsorption sites of containers in two different ways: Treatment of the surface with saturating agents either before or during the stability study. Based on available data, two saturating agents were selected: bovine serum albumin and tween 20 [48,55,56]. These assays were not conclusive, therefore, the adsorption hypothesis was rejected. Pharmaceutical insulin solutions contain zinc in order to physically stabilise the polypeptide without any chemical impact [24]. These ions (also contained in the trace element solution added to the PNA), are used in pharmaceutical formulations to provoke the hexamerisation of insulin, particularly when phenol or metacresol are also present (similarly to in Umuline rapide^TM^) [13]. In such a case, we hypothesised that epitopes on the surface of insulin are hidden in the core of the hexamer and so are inaccessible for antibody recognition. This hypothesis was rejected, but the absence of any difference can be explained: the low concentration of insulin would limit hexamerisation (the monomeric form is the main form at a low concentration). Considering that insulin was unstable in three-in-one PNA, assays were conducted progressively: the influence of each macronutrient (dextrose, amino acids and lipids) was tested. Moreover, each lipid emulsion used in PNA can be composed of one or several oils, coming either from animals or plants, therefore, we hypothesised that the nature and concentration of lipid emulsion could have an impact on the stability of insulin. Nevertheless, neither the nature nor concentration was statistically significant. Lipid peroxydation (leading to the synthesis of malondialdehyde) is a well-known chemical reaction that has already been described in three-in-one PNA [57,58,59]. Consequently, it could explain insulin instability. A stability study was performed with an excessive addition of malondialdehyde (data not shown). There was no conclusive result, leading to this hypothesis also being rejected.

The influence of pH was studied at values 3.6 and 6.3, as insulin stability was assessed in dextrose (pH 3.6) and in the amino acid chamber (pH 6.5) of a PN bag; insulin instability on the other hand was assessed in a two-in-one PNA (pH 6.3). Insulin concentration stability was reported in all PN media at pH 3.6. The ANCOVA analysis confirmed the statistical significance of dextrose and time, both alone and in interaction with pH (an increase in each factor intensified the insulin instability). However, the amino acid content in interaction with pH is inversely related to insulin instability (indeed, it was the only tested medium in which insulin was stable at pH > 6).

It is obvious that artificially decreasing the pH of PN admixtures could be beneficial in improving insulin stability. Nevertheless, considering that amino acid solutions buffer the entire admixture at a pH of around 6–7, it would be difficult to obtain an acidic pH. Moreover, this study was performed in vitro, and has not been assessed in a clinical context; thus, the issue of metabolic acidosis has not been addressed.

Other experiments must also be performed to accurately identify the mechanism of the decrease in insulin content over time. Indeed, the present study used an immunoassay, involving antigen–antibody recognition. If we consider that a chemical reaction could provoke a conformational modification, decreasing this recognition step, it is not possible to predict a clinical or metabolic consequence. Thus, in vivo experiments or other analytical tools will be required to assess the maintenance or loss of the hypoglycaemic effect of insulin after addition to a PN admixture.

Based on these results, it seems that the reaction causing insulin instability is mainly linked to the presence of dextrose, pH value (around 6.3) and time. The higher the glucose content, the greater the reaction. An interaction could occur between insulin and dextrose contained in the same PN medium. Indeed, the glycation of bovine, porcine and human insulin has already been described with proportionality between glycation, the dextrose concentration and time of contact [60,61,62]. Nevertheless, other experiments must be performed because, in the literature, glycation is forced under reducing conditions [61,62,63,64], which was not relevant in our work. High performance liquid chromatography coupled with mass spectrometry could be a method of choice to identify and quantify the glycated insulin in PNA.

The ANCOVA results suggest that other, as yet unidentified, factors must occur in order to fully explain insulin instability in PN. In particular, the role of lipids must be highlighted. Aggregation, which occurs at neutral pH and leads to high molecular weight degradation products of insulin, must be studied [65,66]. The main advantage of this study is that results can be generalised to most PNAs.

## 5. Conclusions

The addition of insulin to two-in-one and three-in-one PNA seems to provoke in vitro insulin instability: its concentration decreases by almost 40% in 8 h when PN contains dextrose at a pH of around 6.3. Glycation, which has already been described for insulin but under reducing conditions, could explain this phenomenon, however, this must be confirmed by further experiments.

## Figures and Tables

**Figure 1 pharmaceutics-13-00458-f001:**
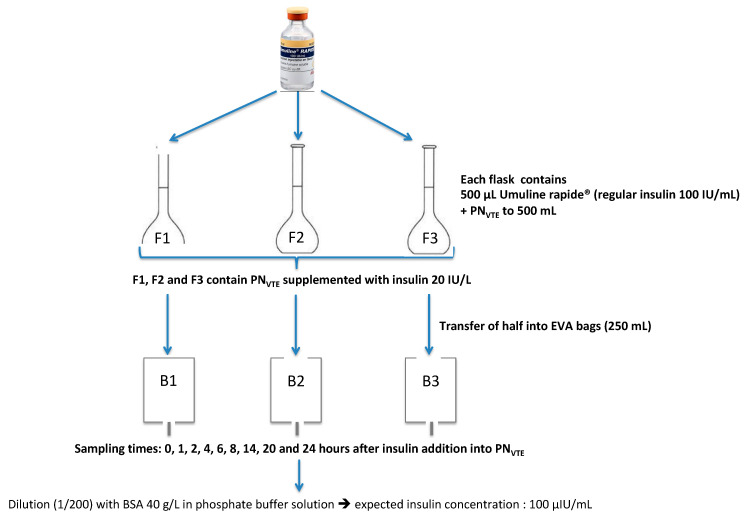
Stability study design. PN_VTE_: parenteral nutrition supplemented with vitamins and trace elements; F: glass flask; B: Ethylene Vinyl Acetate (EVA) bag; BSA: bovine serum albumin. Addition of insulin at a final concentration of 20 IU/L in different PN media. Final PN admixture containing insulin was divided into two equal volumes in order to perform the assay in both glass flasks and EVA bags.

**Figure 2 pharmaceutics-13-00458-f002:**
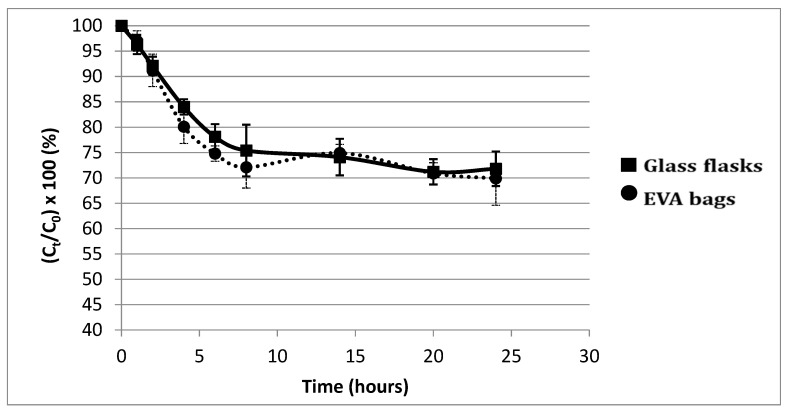
Evolution of insulin concentration (C_t_) as a percentage of initial insulin concentration (C_0_) in 3-in-1 PN admixture in glass flasks and EVA bags over 24 h. Insulin concentration sharply decreased, regardless of the nature of the container materials.

**Figure 3 pharmaceutics-13-00458-f003:**
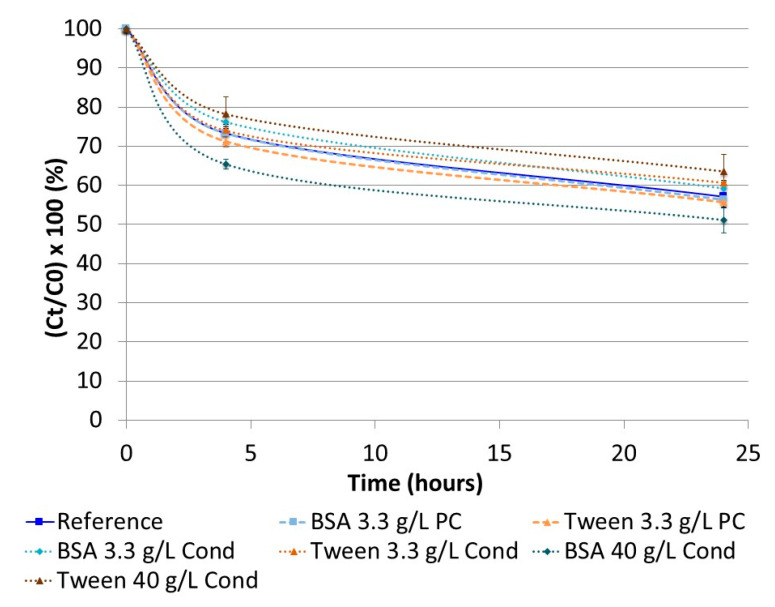
Evolution of insulin concentration during preconditioning and conditioning assays. Reference: 3-in-1 PNA with no preconditioning or conditioning; BSA: bovine serum albumin; PC: preconditioning; Cond: conditioning. Preconditioning and conditioning have no impact on insulin stability in a ternary PN admixture.

**Figure 4 pharmaceutics-13-00458-f004:**
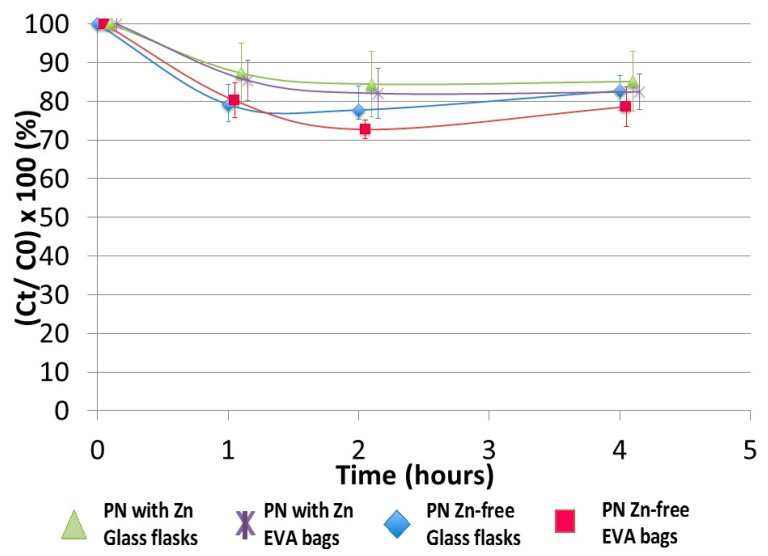
Impact of presence or absence of zinc in ternary PN admixture on insulin concentration. Zinc has no impact on insulin stability in ternary PN admixture.

**Figure 5 pharmaceutics-13-00458-f005:**
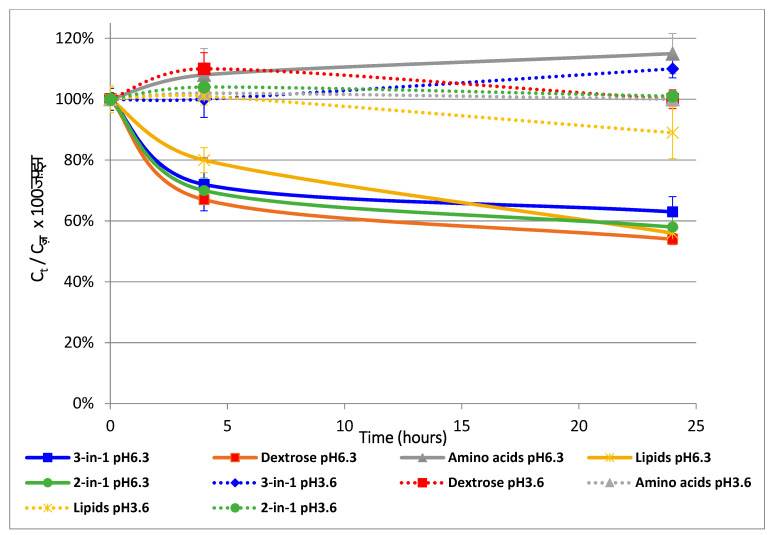
Changing of insulin content in PN admixtures at various pH over time. Continuous line and dotted line correspond, respectively, to pH 6.3 and 3.6. Insulin concentration remained stable at pH 6.3, whereas it decreased at pH 3.6, except in the amino acid solution.

**Figure 6 pharmaceutics-13-00458-f006:**
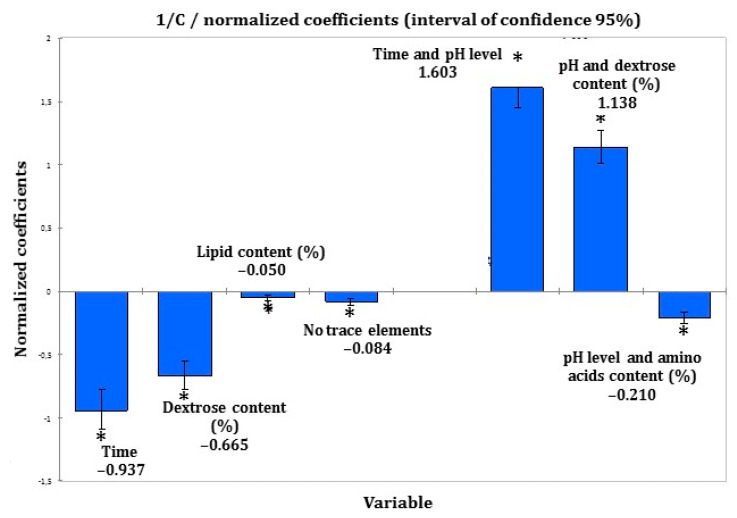
ANCOVA normalised coefficients (CI 95%); *: *p* < 0.0001.

**Table 1 pharmaceutics-13-00458-t001:** Composition and pH values of PN media compounded from an Olimel^TM^ N7E bag Dextrose 35% solution + calcium (A), amino acids 11.1% solution + electrolytes except calcium (B) and lipid emulsion 20% (C).

Media	Composition	pH
Ternary (3-in-1)	A + B + C	6.34
Binary (2-in-1)	A + B	6.34
Dextrose	A	3.6
Amino acids	B	6.5
Lipids	C	>7.0

**Table 2 pharmaceutics-13-00458-t002:** Nature and concentration of lipid emulsions used for stability study.

Brand Name	Nature of Lipids (Oil Origin)	Concentrations (%)
Olimel^TM^ lipid emulsion	Olive oil (80%) + soybean oil (20%)	5, 10, 20
Medialipide^TM^	Soybean oil (50%) + medium-chain fatty acids (50%)	20
Intralipide^TM^	Soybean oil	20
Omegaven^TM^	Fish oil	10
Smoflipid^TM^	Medium-chain fatty acids (30%) + soybean oil (30%) + olive oil (25%) + fish oil (15%)	20

**Table 3 pharmaceutics-13-00458-t003:** Factors taken into account for ANCOVA analysis.

Quantitative Data	Qualitative Data
1/C_t_	Trace elements
Time	Zn
pH level (3.5 or 6.5)	Pre conditioning
Dextrose content (%)	Conditioning
Amino acid content (%)	Nature of lipid emulsion
Lipid content (%)	Container

**Table 4 pharmaceutics-13-00458-t004:** Factors used in ANCOVA.

Independent Variables	Correspondence with Equation (2)	Variables in Interaction	Correspondence with Equation (2)
Time (hours)	A	Time and pH level	E
Dextrose content (%)	B	pH level and dextrose content	F
Lipid content (%)	C	pH level and amino acid content	G
Absence of trace-elements	D		

## Data Availability

Data are available upon request to the corresponding author.

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
