# Peer review of "Addition of Regular Insulin to Ternary Parenteral Nutrition: A Stability Study"

_pharmaceutics, 2021, doi:10.3390/pharmaceutics13040458_

Round 1

Reviewer 1 Report

Manuscript Pharmaceutics-1148507, Henry, H. et al., "Addition of human insulin to ternary parenteral nutrition: a 2 stability study "

In this manuscript, the authors tested the stability of fast-acting human insulin (20 IU/L) in different parenteral nutrition admixtures during a 24 hour incubation in 2 types of containers, glass flasks and EVA bags. Using an electrochemiluminescence immune-assay, insulin concentration was measured and the influence of different factors, time, pH, presence or absence of dextrose, Zinc, amino acids or the saturation of adsorption sites using BSA or tween. The authors used ANCOVA to determine the influence of the different factors and suggest that the addition of insulin to PN decreases the stability of insulin with time, particularly in the presence of dextrose and at the pH of 6.3.

Overall, this is a well written manuscript, I have very few concerns.

  • The authors should discuss the stability of other insulins, such as the regular and long-lasting insulins, in PN and it might compare to fast acting insulin.

  • The authors made the observation that lower acidic pH results in the stabilization of insulin. Do the authors believe that deceasing the pH of PN admixtures will be beneficial? If so, what is their response to the issue of metabolic acidosis? This could perhaps be added to the discussion.

  • Figure 4: legend is a bit fuzzy, we can’t read it clearly.

  • The authors should add little more descriptive details in the figure legends.

Reviewer 2 Report

These are my general/specific comments.

In their paper entitled “Addition of human insulin to ternary parenteral nutrition: a stability study”, the authorsaimed to evaluate the stability of insulin after addition into a ternary PN admixture and to identify influencing factors.  

Comments:

  1. The authors should review the English throughout the paper.
  2. In their abstract, the authors should state the abbreviation of PN (Parenteral nutrition)
  3. In their Abstract/Introduction it is recommended that the authors state an working hypothesis.
  4. In the discussion section authors should consider discussing in more detail the mechanism behind the association of in vitro stability of human insulin in a 3-in-1 PN admixture with metabolic diseases.

Reviewer 3 Report

The authors of the article "Addition of human insulin to ternary parenteral nutrition: a  stability study" presented a complex analysis of insulin's stability added to ternary parenteral nutrition. The advantage of the research is to investigate the influence of many various factors on the stability of insulin (container type, environment, type of fat emulsion, pH, presence of zinc, BSA, and Tween 20).

The writing is fluent and easy to understand. The discussion of the results may be of interest not only to parenteral nutrition clinicians.

I have a few comments/questions:

  • Line 87: "previously validated for insulin quantification" - What parameters have been validated? What is the LOD and LOQ of this method?
  • Line 161: Should this notation be under Figure 1 (not under Figure 2)?
  • The legend in Figure 4 is illegible.
  • Are the data in Figure 5 correct? How do you explain the insulin content of ~ 117% after 24 hours of storage in amino acids with a pH of 6.3?
  • Minor editorial errors, such as missing space, should be corrected (e.g., 0.04M, 1.5L, 3.3g / L, pH3.6).

Round 2

Reviewer 2 Report

These are my general/specific comments.

In their paper entitled “Addition of human insulin to ternary parenteral nutrition: a stability study”, the authorsaimed to evaluate the stability of insulin after addition into a ternary PN admixture and to identify influencing factors.  

Comments:

  1. The authors improved the English throughout the paper.
  2. The authors stated the abbreviation of PN (Parenteral nutrition) and also the working hypothesis.
  3. The authors improved the discussion section.

This manuscript is a resubmission of an earlier submission. The following is a list of the peer review reports and author responses from that submission.